# Leishmania RNA virus-1 is similarly detected among metastatic and non-metastatic phenotypes in a prospective cohort of American Tegumentary Leishmaniasis

Braulio Mark Valencia[1]*, Rachel Lau[2], Ruwandi Kariyawasam[3], Marlene Jara[4], Ana Pilar Ramos[5,6], Mathilde Chantry[7], Justin T. Lana[8], Andrea K. Boggild[3,9,10†], Alejandro Llanos-Cuentas[5,6†]

1 Kirby Institute, The University of New South Wales (UNSW Sydney), Sydney, Australia, 2 Public Health Ontario Laboratory, Toronto, Canada, 3 Institute of Medical Science, University of Toronto, Toronto, Canada, 4 Institute of Tropical Medicine of Antwerp, Antwerp, Belgium, 5 Instituto de Medicina Tropical "Alexander von Humboldt", Universidad Peruana Cayetano Heredia, Lima, Peru, 6 Hospital Nacional Cayetano Heredia, Lima, Peru, 7 CER Groupe, Aye, Belgium, 8 Nicholas School of the Environment, Duke University, Durham, North Carolina, United States of America, 9 Tropical Disease Unit, Toronto General Hospital, Toronto, Canada, 10 Department of Medicine, University of Toronto, Toronto, Canada

† These authors contributed equally to this work.
* barroyo@kirby.unsw.edu.au

**Data Availability Statement:** All relevant data are within the manuscript and its Supporting

## Abstract

American Tegumentary Leishmaniasis (ATL) is an endemic and neglected disease of South America. Here, mucosal leishmaniasis (ML) disproportionately affects up to 20% of subjects with current or previous localised cutaneous leishmaniasis (LCL). Preclinical and clinical reports have implicated the Leishmania RNA virus-1 (LRV1) as a possible determinant of progression to ML and other severe manifestations such as extensive cutaneous and mucosal disease and treatment failure and relapse. However, these associations were not consistently found in other observational studies and are exclusively based on cross-sectional designs. In the present study, 56 subjects with confirmed ATL were assessed and followed out for 24-months post-treatment. Lesion biopsy specimens were processed for molecular detection and quantification of *Leishmania* parasites, species identification, and LRV1 detection. Among individuals presenting LRV1 positive lesions, 40% harboured metastatic phenotypes; comparatively 58.1% of patients with LRV1 negative lesions harboured metastatic phenotypes ($p = 0.299$). We found treatment failure ($p = 0.575$) and frequency of severe metastatic phenotypes ($p = 0.667$) to be similarly independent of the LRV1. Parasite loads did not differ according to the LRV1 status ($p = 0.330$), nor did Leishmanin skin induration size ($p = 0.907$) or histopathologic patterns ($p = 0.780$). This study did not find clinical, parasitological, or immunological evidence supporting the hypothesis that LRV1 is a significant determinant of the pathobiology of ATL.

Information files. (complete dataset has been included as S1 Dataset).

**Funding:** This project was funded by an International Society for Infectious Diseases (ISID) small grant (non-numbered grant). BMV is supported by a Scientia PhD Scholarship from UNSW Sydney (non-numbered grant). The funders had no role in study design, data collection and analysis, decision to publish, or preparation of the manuscript.

**Competing interests:** The authors have declared that no competing interests exist.

## Author summary

The Leishmania RNA virus-1 (LRV1) has been implicated as a possible modulator agent in the pathogenesis of leishmaniasis. In-vivo and in-vitro studies have depicted specific mechanisms of how LRV1 could lead to metastasis. Clinical studies and epidemiological evidence have both supported and rejected the hypothesis that LRV1 is a relevant determinant of progression, treatment failure and clinical severity of American Tegumentary Leishmaniasis (ATL). This lack of consistency between preclinical and clinical reports requires further longitudinal studies to clarify the role of LRV1 in ATL. Due to the complex nature of ATL, as other frequent human diseases, these studies should tackle multiple determinants of pathogenicity, including LRV1 status, parasite features, immune status, and prevalent comorbidities affecting individuals in endemic settings. Also, critical methodological aspects allowing for the reliable identification and quantification of LRV1 should be guaranteed.

## Introduction

American Tegumentary Leishmaniasis (ATL), a neglected tropical disease predominantly caused by New World *Leishmania Viannia* pathogens, represents almost one-third of the global cutaneous leishmaniasis burden [1]. *L. (V.)* pathogens are more prone to cause mucosal leishmaniasis (ML), the metastatic phenotype causing destructive and debilitating mucosal lesions in 5–20% of individuals previously or concurrently affected by localised cutaneous leishmaniasis (LCL) [2,3]. Although ML has been mainly associated with *L. (V.) braziliensis* infections [4–6], its occurrence is being increasingly documented among other *L. (V.)* pathogens [7–9] as well as the Old World parasites *L. (L.) major, L. (L.) infantum, and L. (L.) Donovani* [10–13]. It is unclear if the emergence of the non-*L.(V.) braziliensis* ML cases have occurred because of changes in pathobiological mechanisms, improved epidemiological reports, or improved access to species identification through molecular biology. Independently of the infecting pathogen, the predominance of LCL cases over ML—or even the less common phenotypes of disseminated leishmaniasis (DL) or diffuse cutaneous leishmaniasis (DCL)—suggest that pathogen determinants are not solely explaining the diversity of clinical manifestations and phenotypes. The predominance of subclinical or mild manifestations over severe or unusual phenotypes is frequently reported in many other infectious diseases where unusual phenotypes represent a lower proportion of the disease burden [14].

The LRV1 has been implicated as a possible determinant of the pathogenesis of ATL since its presence seems to connect the primary infection with the subsequent development of metastatic and hyper-inflammatory disease [15–17]. DL triggered by LRV1 has been documented in one animal model [17], and clinical reports have shown a high prevalence of LRV1 among ML cases [18]. These findings have, undoubtedly, increased the interest in LRV1 due to its potential role in the metastatic process and its potential applicability in diverse fields as diagnosis, therapeutics, biomarkers or treatment response and severity. To better understand the role LRV1 might play in the dissemination of ATL parasites, association with treatment failure or disease severity, we explored (1) if identification of LRV1 is more frequent in metastatic phenotypes (i.e., ML, MCL or DL) than those with a non-metastatic phenotype (i.e., LCL) (2) if identification of LRV1 is more frequent in patients who experience treatment failure, and (3) if the severity of the metastatic phenotype is associated to the LRV1 status.

## Methods

### Ethics statement

The study received Institutional Review Board approval from both Universidad Peruana Cayetano Heredia and Hospital Cayetano Heredia. All participants provided written informed consent. Minors were not included.

### Study design

Between March 2012 through October 2016, we conducted a prospective cohort study based out of the Leishmaniasis Clinic, Institute of Tropical Medicine Alexander von Humboldt–Hospital Cayetano Heredia, in Lima, Peru. Our study included fifty-six subjects with LCL, ML, or DL confirmed by PCR, smear, or histopathology (Fig 1B–1D). Severe metastatic manifestations were defined as ML extending over the upper and lower respiratory tract or DL

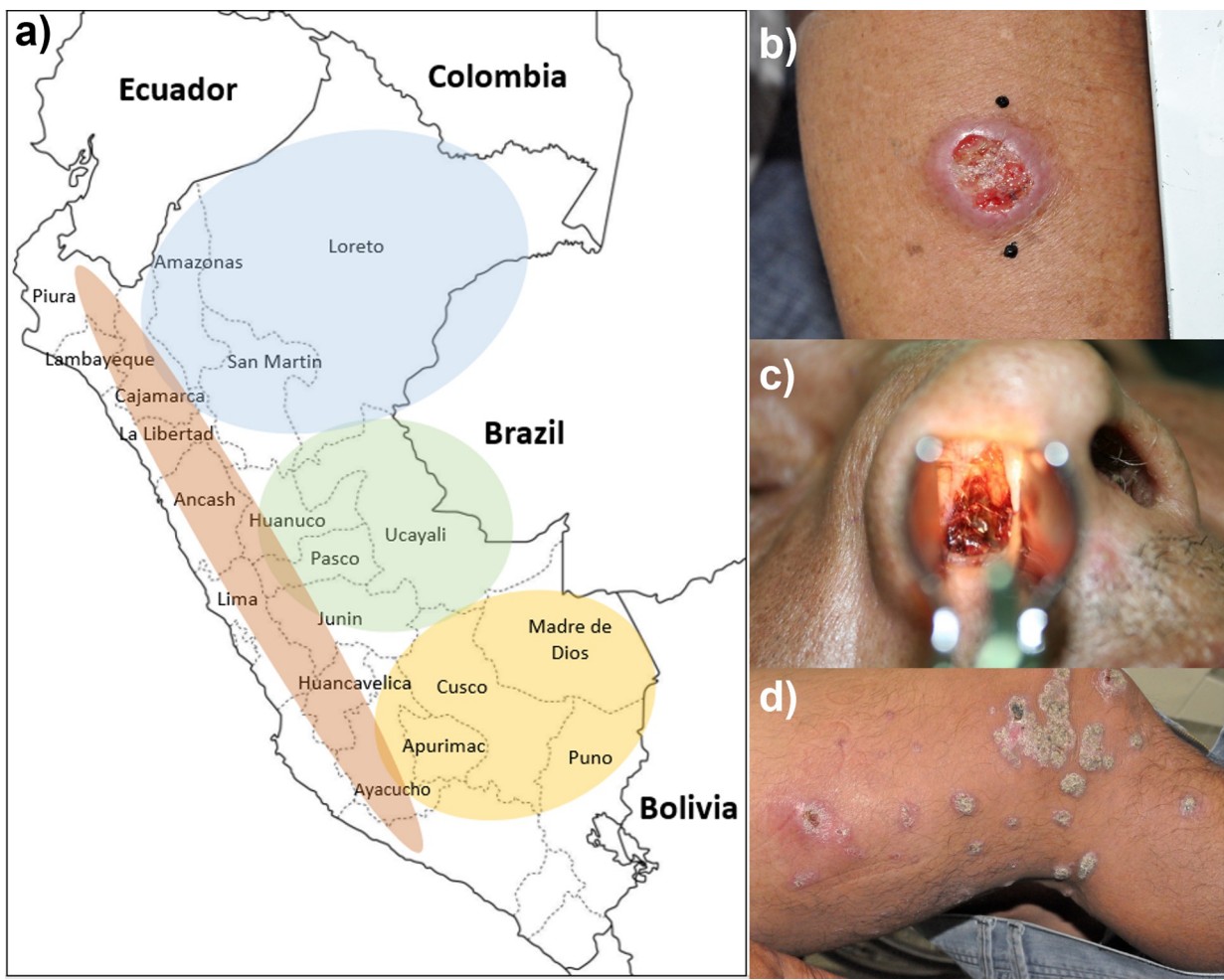

**Fig 1.** A) Regions where ATL patients got infected. They are referred as south jungle (yellow), central jungle (green), north jungle (blue), and highlands (orange) due to previous epidemiologic reports demonstrating the predominance of leishmania species occur in these areas. B) Typical LCL case in a participant without metastatic manifestations. C) Typical ML case in a participant presenting LCL 3 years before current diagnosis. D) DL case in one participant with multiple non-ulcerative lesions involving lower limbs (photo), face, and chest. Fig 1A was obtained from https://colorearimagenes.net/wp-content/uploads/2021/03/MapaPeru36.jpg and the delimitation of regions (yellow, green, blue, orange) and the names of cities and countries were added by BMV. Fig 1B and 1C were taken from patients enrolled in the study by BMV.

refractory to two consecutive treatments. Subjects were enrolled consecutively according to their request for medical care. Exclusion criteria were (1) age under 18 years old, (2) facial LCL, (3) prior systemic therapy during the last three months, or (3) inability to provide written informed consent.

## Treatment and follow-up

Participants received treatment according to standard guidelines and based on physician judgement (BMV, ALLC, APR). Briefly, individuals with LCL received Sodium stibogluconate (SSG) 20mg/Kg/day for 20 days, participants with ML or DL received SSG for 30 days (in the case of ML, nasal +/- oral involvement) or Amphotericin B (AMB) 25mg/Kg (relapsing DL or ML with nasal + oral + larynx involvement). When therapy was finalised, patients were followed-up at months 1–6, 12, and 24 post treatment. After the 24-month follow-up, some individuals requested additional yearly clinical evaluations, which continue today as a part of our standard of care.

## Sampling for diagnostic and investigational procedures

Leishmanin skin test (LST): LSTs were applied using 0.1 mL of in-house, sterile, heat-killed promastigote lysate in 0.005% thimerosal as described elsewhere [19] and read at 48 h after administration. A positive result was indicated by ≥5 mm of erythema and induration [19,20].

Cytology brushes: After removing any overlying scab or crust with sterile tweezers, two sterile and duplicate CerviSoftH cervical cytology brushes (Puritan Medical Products, Maine) were rolled clockwise on mucosal or cutaneous lesions five times each in sequence. The first cytology brush was broken off into a 1.5-mL Eppendorf tube containing 700 mL 100% ethanol and stored at -22˚C for strain identification and parasite load quantification. The second cytology brush was broken off into a 1.8-mL cryovial containing 1000 mL RNAlater, incubated overnight and stored at -22˚C until LRV1 detection assays were performed.

Mucosal or cutaneous biopsies: After the collection of cytology brushes, mucosal or cutaneous lesions were anesthetised. Two small biopsy specimens were then obtained from lesions. In the case of ML, biopsy specimens were obtained using sterile nasal ethmoidal biopsy forceps, whereas LCL specimens were obtained with a 2-mm punch. Tissue was then stored in a 1.8-mL cryovial containing 1000 mL RNA later for qualitative PCR detection of LRV1 RNA or placed in 10% formalin for histopathology with hematoxylin and eosin, Ziehl-Neelsen, and Giemsa staining. Sterile gauze was applied with pressure to cutaneous or mucosal lesions until haemostasis was achieved.

**Species identification and quantification of Leishmania (Viannia) spp. by quantitative real-time PCR.** Samples were centrifuged at 3000 g for 5 min, and ethanol was discarded. Biopsied tissues were disaggregated with a sterile scalpel. According to the manufacturer's instructions, disaggregated tissue and cytology brushes were processed for DNA isolation using the High Pure PCR Template Preparation KitH (Roche, Mannheim, Germany). Leishmania kDNA PCR was performed using primers and conditions described previously [9,21]. Parasites were typed according to a priorly reported algorithm [22]. Specimens unclassifiable by this algorithm were characterised by end-point PCR targeting the internal transcriber space 1 (ITS1), ITS2, cysteine proteinase B, heat shock protein 70, mannose phosphate isomerase, and zinc-dependent metalloproteinase (GP63) [23–26]. Restriction fragment length polymorphism analysis was performed on each product of end-point PCR [23,25,26]. The qPCR assay based on kDNA minicircle amplification (kDNA qPCR) was used to detect and quantify *Leishmania (Viannia)* DNA in biological samples as described by Jara et al. [27]

### LRV detection in clinical samples

RNA isolation and quantification: The lesion biopsy specimens from patients with LCL, ML, and DL were minced with a sterile scalpel and then homogenised with 0.75 mL of Trizol Ls in a Tenbroeck Homogenizer (Bellco). Later, the RNA was isolated according to the manufacturer's instructions. The isolated RNA was quantified by fluorometry using the Qubit fluorometer (Invitrogen).

cDNA synthesis: RNA was treated with the RQ1 RNase-Free DNase (Promega). Then, according to the manufacturer's instructions, 0.2–1.0 μg of RNA (this amount varied according to biopsy specimens) was used for cDNA synthesis with the Transcriptor First Strand cDNA Synthesis Kit (Roche).

RT-qPCR for LRV detection: 5 μL of cDNA (1/10) was added to 25 μl quantitative real-time PCR (qPCR) reactions, which contained 1X iQ SYBR Green Supermix (Bio-Rad) and 300 nM of both primers, namely LRVF 5-GAG TGG GAG TCC CCC ACA T-3, and LRVR 5-TGG ATA CAA CCA GAC GAT TGC T-3 [28]. Reactions were run on the LightCycler 480 system (Roche). The thermal cycling conditions were: 95˚C for 3 min, 44 cycles at 95˚C for 15 s, 60˚C for 20 s and 72˚C for 10 s. Fluorescence emission was measured at the end of the elongation step. After PCR amplification, a melting curve was generated to check the amplicon specificity; it consisted of 1 cycle at 95˚C for 60 s, followed by 60˚C for 60 s and continuous heating at 0.02˚C/s to 95˚C. The PCR product had a specific Tm of 79.37˚C. Each run included a positive control sample (cDNA from promastigotes of the strain MHOM/BR/75/M4147), a negative control (cDNA from peripheral blood mononuclear cells) and a blank (no-template control). Each sample was tested in triplicate.

### Statistical analysis

Descriptive statistics and comparisons of groups (i.e. LRV1 status among phenotype) were performed using Stata/IC version 14 (StataCorp, College Station, Texas, USA). The association of LRV1 status with clinical manifestations was tested in grouping participants by metastatic (ML or DL) and not metastatic phenotypes (LCL). Both groups were compared using the Chi2 or Fisher's exact tests for qualitative variables and Student's 2-tailed t-test or Wilcoxon rank-sum test to compare continuous variables (i.e., age, duration of disease, and the number of lesions). Possible interactions between variables were analysed. Differences were considered significant when p-values were < 0.05.

## Results

### Epidemiological, clinical and parasitological features according to phenotypes

Table 1 summarises these key epidemiological and clinical features, including LRV1 status according to the phenotypes of interest. Participants were predominately males (85.7%). Median age of participants was 39 years (range 20–82) and the medium time in which they had symptoms prior to enrolment was 9.85 months (range 0.7 months—41 years). ATL was acquired in the central jungle (39.3%), south jungle (35.7%), north jungle (16.1%) and highland regions (8.9%) of Peru (Fig 1A). Almost one-third of participants acquired the disease in the same area in which they had lived their entire lives, and thus, we defined them as native cases (27%). Twenty-six participants had LCL (46.4%), whereas 30 had metastatic disease (53.6%). For participants with metastatic disease, the median latency period (from primary cutaneous illness to the development of metastatic disease) was 5.6 years; only 5 of them (16.7%) received standard antimonial therapy for their primary LCL. The diagnosis was confirmed by direct examination in 57.6% of participants, although all were PCR-positive. LST

**Table 1. Clinical and epidemiological characteristics by the phenotype of interest.**

| Characteristic | Metastatic disease (n = 30) | Localised disease (n = 26) | p-value |
|---|---|---|---|
| Male sex (%) | 27 (90) | 21 (80.7) | .325 |
| Median age in years (IQR) | 44 (34–54) | 30.5 (26–46) | .014 |
| No-native from endemic region (%) | 25 (83.3) | 16 (61.5) | .066 |
| Median duration of symptoms in weeks, (IQR) | 31.5 (12.8–80) | 4.4 (2.6–5.3) | < .001 |
| Infecting agent, (%) | | | .110 |
| *L. (V.) braziliensis*<br> *L. (V.) peruviana*<br> *L. (V.) guyanensis*<br> Non identifiable | 14 (46.7)<br>3 (10)<br>2 (6.6)<br>11 (36.7) | 13 (50)<br>2 (7.7)<br>7 (26.9)<br>4 (15.4) | |
| Mean Parasite load$^£$ | 4.86 ± 3.2 | 6.8 ± 4.7 | .097 |
| Positive smear (%) | 3 (37.5) | 16 (64) | .238 |
| Positive LST (%) | 25 (83.3) | 24 (92.3) | .431 |
| Mean LST induration in mm | 12.2 ± 7.7 | 11.4 ± 6.4 | .672 |
| LRV1 positive | 4 (13.3) | 6 (23.1) | .450 |
| Relevant comorbidities$^*$ | 2 (6.7) | - | .18 |

$^£$ Parasite load values were log-transformed due to the scattering values.

$^*$ Participant 1 was diagnosed with HIV due to recent onset DL. Participant 2 was diagnosed with multisystemic tuberculosis without HIV infection.

was positive in 86% of cases with a mean induration of 11.8 ± 7 mm. By histopathology, patients were characterised as having well-circumscribed granulomas (37.5%), plasmacytic infiltrates (PI) without evidence of granulomas (17.9%) and giant/epithelioid cells and discrete PI (44.6%). After therapy, patients were followed at least for 24 months, although some were assessed for more extended periods (median follow-up 42.7 months). At the end of this period, no patient with LCL developed metastatic manifestations. Only age and time since symptoms onset differed among metastatic and localised diseases.

## Influence of the LRV1 in the severity of clinical manifestations and immunopathology

Table 2 describes the diversity of metastatic phenotypes and their clinical and immunopathologic severity according to the LRV1 status. Among LRV1 positive cases, 4 of 10 (40%) individuals presented with metastatic phenotypes, whereas 25 of 43 (58%) of LRV1 negative patients presented with metastatic phenotypes (*p* = 0.299). ML was the predominant metastatic manifestation in both groups. Unusual phenotypes expected to be associated with LRV1 such as DL or simultaneous mucocutaneous involvement (mucocutaneous leishmaniasis, MCL) were all LRV1-. None of the metastatic/LRV1+ patients received standard antimonial therapy for the primary LCL, while only 5 (16.7%) of the metastatic/LRV1- were treated correspondingly. Severe metastatic manifestations were reported in one LRV1+ case and 9 LRV1- cases. Among these ten severe cases, only 2 received conventional therapy for the primary LCL. Duration of symptoms, infecting agent, parasite load, the intensity of the LST, and histopathologic characterisation were also similar across the LRV1 positive and negative groups. Interestingly, LRV1 was reported in *L.(V) peruviana* for the first time in a case of ML.

## Treatment response according to LRV1 status

As shown in Table 3, 35 participants cured their disease definitively (62.5%) while 21 individuals experienced relapses and required consecutive therapies. Factors typically associated with

**Table 2. Spectrum of clinical manifestations and their severity according to the LRV1 status.**

| Characteristic | positive (n = 10) | negative (n = 43) | p-value |
|---|---|---|---|
| Metastatic phenotype*, (%) | | | .554 |
| ML | 4 (100) | 16 (64) | |
| PML | - | 3 (12) | |
| MCL | - | 3 (12) | |
| DL | - | 3 (12) | |
| Severe Metastatic phenotype&, (%) | 1 (25) | 9 (36) | .667 |
| Median duration of symptoms in weeks, (IQR) | 7.2 (2.8–60) | 10 (3.9–34.4) | .657 |
| Infecting agent, (%) | | | .187 |
| - L. (V.) braziliensis | 5 (50) | 22 (51.1) | |
| - L. (V.) peruviana | 1 (10) | 4 (9.3) | |
| - L. (V.) guyanensis | 2 (20) | 7 (16.3) | |
| - Non identifiable | 2 (20) | 10 (23.3) | |
| Mean Parasite load£ | 6.9 ± 4.71 | 5.5 ± 3.93 | .330 |
| Mean LST induration in mm | 11.7 ± 3.9 | 12 ± 7.8 | .907 |
| Predominant histologic pattern | | | .787 |
| Plasmocytic infiltration (PI) | 1 (10) | 8 (18.6) | |
| Giant/epithelioid cells + PI | 5 (50) | 18 (41.9) | |
| Well circumscribed granulomas | 4 (40) | 17 (39.6) | |

& ML extending over the upper and lower respiratory tract or DL refractory to two consecutive treatments

£ Parasite load values were log-transformed due to the scattering values.

treatment response (i.e. infecting agent, chronicity of lesions, size of LST induration, and local immunopathological reactions) were not related to therapeutic outcomes in this population. As in other phenotypes, the LRV1 status was also not associated with treatment response (p = 0.575). Baseline parasite load, a potential treatment response biomarker, was similar among relapsing and resolved cases.

**Table 3. Factors associated with treatment response.**

| Characteristic | Failure (n = 21) | Cured (n = 35) | p-value |
|---|---|---|---|
| Male sex (%) | 19 (90) | 29 (82.8) | .430 |
| Median age in years (IQR) | 36 (28–53) | 40 (29–50) | .767 |
| Median duration of symptoms in weeks, (IQR) | 7.8 (3.2–38.3) | 12.2 (4.2–28.7) | .932 |
| Infecting agent, (%) | | | .397 |
| L. (V.) braziliensis | 13 (61.9) | 14 (40) | |
| L. (V.) peruviana | 2 (9.5) | 3 (8.6) | |
| L. (V.) guyanensis | 2 (9.5) | 7 (20) | |
| Non identifiable | 4 (19.1) | 11 (31.4) | |
| LRV1 positive | 3 (15) | 7 (21.2) | .575 |
| Mean Parasite load£ | 6.8 ± 4.8 | 5.2 ± 3.61 | .206 |
| Mean LST induration in mm | 11.8 ± 6.8 | 11.8 ± 7.2 | .969 |
| Predominant histologic pattern | | | .710 |
| Plasmocytic infiltration (PI) | 3 (14.3) | 7 (20) | |
| Giant/epithelioid cells + PI | 10 (47.6) | 15 (42.8) | |
| Well circumscribed granulomas | 8 (38.1) | 13 (37.1) | |

£ Parasite load values were log-transformed due to the scattering values.

## Discussion

Despite more evidence on the LRV1 is available, its role in the pathobiology of ATL keeps unclear, and evidence is still contradictory. We found no overrepresentation of LRV1 among patients with metastatic disease, nor did we find an association between LRV1 positivity and treatment failure or disease severity. Contrarily to previous reports limited by a cross-sectional design, our findings are the first to be based on a prospective evaluation, which unlike previous reports, were able to simultaneously assess other potential determinants of disease severity such as local parasite burden, local and systemic immune responses, or concurrent medical conditions with a significant impact on clinical phenotypes. To the best of our knowledge, we report the first LRV1 infection in *L.(V) peruviana*.

As noted, other observational studies have reported contradictory findings, which may be in part due to their cross-sectional nature. In terms of progression to ML, LRV1 was reported more frequently in ML cases [18,29,30], and similarly or less prevalent in ML cases [31–33]. It has been shown be associated with an increased risk of treatment failure [32,34]; and, it has been shown not to be related to detrimental treatment outcomes [33,35]. Furthermore, LRV1 has also been associated with extensive disease, although this finding only relied on ML [29]. A systematic review and meta-analysis involving Old and New World pathogens concluded LRV is more prevalent among ATL however, the same review found no significant difference comparing metastatic (36.3%) versus non-metastatic phenotypes (32.3%) [36]. As highlighted in this systematic review, bias in diagnostic methods reliably detecting LRV1 and a limited clinical characterisation of other risk factors were major limitations in all studies. The lack of a prospective assessment, to ensure metastatic or non-metastatic phenotypes are definitive or not, has never been addressed despite the fact that progression from LCL to ML generally takes months or even years after the primary LCL infection has healed [8,37,38]. From that perspective, we removed some uncertainty as to whether phenotypes are correctly representing the contribution of LRV1 to the pathobiology of ATL in this cohort. In terms of treatment response, outcomes cannot be assessed cross-sectionally, and then, follow-up assessments of 3–6 months are highly recommended for LCL [39,40] and at least 24 months for ML or DL [41,42]. For that reason, treatment response is another phenotype lacking appropriated characterisation in previous studies and consequently under or overestimating the contribution of LRV1. Our findings support epidemiologic evidence from French Guiana where, despite the high prevalence of LRV1 infection in *L. (V.) guyanensis*, ML or other metastatic manifestations are very infrequent compared to other ATL endemic countries [43] and treatment failure of LCL was not associated to LRV1[35].

Our study explored multiple factors affecting ATL progression and severity in a cohort of individuals with a consistent phenotype characterisation, including demographics, history of previous treatment, comorbidities, and parasite features. Factors associated with the progression to ML are poorly understood but include age [44], infecting species [4–6], location and extension of the primary LCL and appropriate treatment [8,37,45], impaired immunity [44], and genetic susceptibility [46]. In this cohort, the LRV1 status was similar among metastatic and localised diseases, and only increased age was associated with the development of metastatic disease. It is also important to highlight that DL cases, where a high burden of LRV1 should be expected, were mostly negative and simultaneously harbour immunosuppressive conditions. A trend of more frequency of metastatic manifestations among migrants (non-natives) displaced to endemic settings was also noticed as previously reported in Bolivia and potentially explained by genetic and environmental factors [47–49]. Interestingly, Bolivia and Peru report most ATL cases among migrants with Quechua or Aymara backgrounds, suggesting a significant contribution of genetic and environmental determinants for disease

progression [47,48]. Both countries, reporting the higher incidence of ML globally [1], share this epidemiologic scenario and reported no association of LRV1 with the progression to ML [32].

Preclinical studies are the main body of research arguing LRV1 has a role in the pathogenesis of ATL. Using in-vivo and in-vitro experiments, Ives et al., found early metastatic lesions were characterised by highly LRV1 infected *L. (V.) guyanensis* triggering exacerbated toll-like receptor-3 (TLR-3) mediated inflammatory responses[17]. Similarly, Atayde et al. demonstrated that newly LRV1 infected *L. (V.) guyanensis* parasites tended to generate more severe lesions in an animal model[50]. Here, exosomes seemed to be an essential pathway for LRV1 to infect leishmania parasites and enhance parasite replication and infectivity [50]. LRV1 infected parasites also showed to promote macrophage survival through overexpression of microRNA-155 (miR-155) [51], a small non-coding RNA induced by TLR-3 in response to a broad diversity of pathogen-associated molecular patterns (PAMPs) but also unspecific damage-associated molecular patterns (DAMPs) as type I IFNs, TNF-a, IL-1b, hypoxia [52]. An exacerbated inflammation and increased risk of relapse and dissemination after intralesional administration of type 1 IFNs was similarly identified in a murine model induced by *L. (V.) guyanensis* LRV1+ [53]. Furthermore, LRV1 was demonstrated to evade other pattern-recognition receptors (PRRs) as the RIG-like receptors (RLR) and NOD-like receptors (NLR), confirming the preponderance of TLR-3 and type I IFN responses in the pathobiology of ATL but also supporting their theoretical contribution to the severity of disease [54]. Interestingly, Kariyawasam et al. demonstrated the hijacking of the TLR-3 pathway and a low expression of NLR-triggered cytokines also happens in *L. (V.) braziliensis*[23]. Moreover, inhibition of LRV1 replication in *L. (V.) braziliensis* and *L. (V.) guyanensis* using long-hairpin/stem-loop RNA abolished TLR-3 dependent responses in an in-vitro model [55]. This disbalance of TLR-3 and NLRP3 inflammasome responses was evidenced in ATL patients, where LRV1+ cases displayed lower levels of IL-1β and active Casp1 but similar TNF-α production compared to LRV1 negative [30]. However, these differences could be also consequence of the genetic diversity of genes involved in the NLRP3 inflammasome pathway, and interestingly also associated with the progression and severity to diverse infectious diseases [56–58].

Our research shares significant limitations with previous observational studies. As an RNA virus, the LRV1 life cycle is eminently in the cytoplasm making its presence and activity in amastigotes heterogeneous as is the distribution of amastigotes themselves in lesions. Several studies have demonstrated that the bottoms of ulcers have a higher parasite density than indurated borders [59–62]. In contrast, the distribution of parasites in ML is entirely unknown. Regardless, samples are usually taken from indurated borders, and consequently, a representative collection of infected parasites is not guaranteed. This could be another methodological limitation explaining the diverse prevalence of LRV1 reported in different settings. Additionally, a recent study has shown clinical samples are more reliable than parasite isolates for quantitative characterisation of LRV1, whereas less reliable for qualitative identification[33]. Unfortunately, parasite isolation is feasible in less than 60% of LCL cases, and it is even much lower in ML and other chronic manifestations [63]. Obtaining samples for parasite isolation is an excruciating procedure, making its use for routine or investigational purposes controversial. In this context, samples for reliable LRV1 detection must be taken from the entire lesion or ideally from parts with the highest parasite concentration. Taking several samples from the entire lesion from each single lesion is a potential solution, though this is more likely with LCL as DL and ML present with numerous or more extensive lesions that may present in nasal, oral, and larynx cavities where obtaining samples requires complex medical instrumentation or expert medical skills. Secondly, although some concordant findings were reported in preclinical and clinical studies, the critical metastatic process discussed above seems to happen

very early in the infection [17,50]. ATL lesions develop full features at least one month after the inoculation [64], making early isolation of parasites or clinical sampling particularly challenging. Lastly, despite our longitudinal assessment, the modest sample size significantly affects our results' representativity and confidence.

The diversity of evidence from preclinical, clinical, and epidemiological studies makes the role of LRV1 in human pathobiology still controversial. Because limitations mentioned above were not addressed in this or previous studies, further prospective studies are required to understand better the role of LRV1 in human diseases. These studies should include the same prospective assessment to accurately characterise phenotypes, collaborative involvement of multiple investigators to understand parasite diversity and obtain robust sample sizes, an accusative search of comorbidities driving the development of metastatic manifestations, suitable sample of lesions to increase the likelihood of LRV1 identification, and an appropriate characterisation of unexplored factors affecting progression as the host genetic make-up.

## Supporting information

**S1 Dataset. Dataset containing all variables (including parasite load).**
(XLSX)

## Acknowledgments

The support of laboratory technicians and nurses of the Instituto de Medicina Tropical "Alexander von Humboldt", Universidad Peruana, and health workers and promotors from rural medical centres in Peru are gratefully acknowledged, as well as the enduring cooperation of the subjects in this research. We also acknowledge participants of the first International IUIS-FAIS-Immuno-Ethiopia course for their valuable assessment and feedback on preliminary results of this study.

## Author Contributions

**Conceptualization:** Braulio Mark Valencia, Andrea K. Boggild, Alejandro Llanos-Cuentas.

**Data curation:** Braulio Mark Valencia.

**Formal analysis:** Braulio Mark Valencia.

**Funding acquisition:** Braulio Mark Valencia.

**Investigation:** Braulio Mark Valencia, Rachel Lau, Ruwandi Kariyawasam, Marlene Jara, Ana Pilar Ramos, Mathilde Chantry, Justin T. Lana, Andrea K. Boggild, Alejandro Llanos-Cuentas.

**Methodology:** Braulio Mark Valencia, Rachel Lau, Ruwandi Kariyawasam, Andrea K. Boggild, Alejandro Llanos-Cuentas.

**Resources:** Andrea K. Boggild, Alejandro Llanos-Cuentas.

**Supervision:** Andrea K. Boggild, Alejandro Llanos-Cuentas.

**Writing – original draft:** Braulio Mark Valencia, Justin T. Lana, Andrea K. Boggild, Alejandro Llanos-Cuentas.

**Writing – review & editing:** Braulio Mark Valencia, Rachel Lau, Ruwandi Kariyawasam, Marlene Jara, Ana Pilar Ramos, Mathilde Chantry, Justin T. Lana, Andrea K. Boggild, Alejandro Llanos-Cuentas.

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
