## [Decision Letter · Decision Letter 0]

21 Dec 2021

Dear Dr Valencia,

Thank you very much for submitting your manuscript "Leishmania RNA virus-1 is similarly detected among metastatic and non-metastatic phenotypes in a prospective cohort of American Tegumentary Leishmaniasis" for consideration at PLOS Neglected Tropical Diseases. As with all papers reviewed by the journal, your manuscript was reviewed by members of the editorial board and by several independent reviewers. The reviewers appreciated the attention to an important topic. Based on the reviews, we are likely to accept this manuscript for publication, providing that you modify the manuscript according to the review recommendations. 

Sincerely,

Walderez O. Dutra, PhD.

Deputy Editor

Walderez Dutra

Deputy Editor

Reviewer's Responses to Questions

**Key Review Criteria Required for Acceptance?**

**Methods**

-Are the objectives of the study clearly articulated with a clear testable hypothesis stated?

-Is the study design appropriate to address the stated objectives?

-Is the population clearly described and appropriate for the hypothesis being tested?

-Is the sample size sufficient to ensure adequate power to address the hypothesis being tested?

-Were correct statistical analysis used to support conclusions?

-Are there concerns about ethical or regulatory requirements being met?

Reviewer #1: This is a very well performed study that brings new insights and novel information to the LRV field. The patient cohort was well designed and followed, a distinguished feature of the aforementioned manuscript.

Reviewer #2: The objectives of the study is clear presented. The methods applied were adequate, although better protocols for parasite quantification are already available employing endogenous and exogenous controls.Considering that the Leishmania species might be related to the final prognosis of ATL, to use all them together it is maybe not appropriated. So. I suggest to perform the analysis using L. braziliensis only, as the majority of cases were infected by this species.

**Results**

-Does the analysis presented match the analysis plan?

-Are the results clearly and completely presented?

-Are the figures (Tables, Images) of sufficient quality for clarity?

Reviewer #1: Yes, the way the authors display the data is very clear, concise and objective. Statistical analysis was well-performed. Tables are in accordance with the proposed hypothesis.

Reviewer #2: The results were well presented, but I think that data showing parasite quantification should be presented, at least as supplementary material. Viral load can be an aspect also related to the severity of the disease and unfortunately this was not tested.

**Conclusions**

-Are the conclusions supported by the data presented?

-Are the limitations of analysis clearly described?

-Do the authors discuss how these data can be helpful to advance our understanding of the topic under study?

-Is public health relevance addressed?

Reviewer #1: Yes, the paper is very good and challenges in a very sophisticated way what has been published regarding the correlation between LRV and ML development. For sure, this manuscript deserves publication in PNTD

Reviewer #2: The conclusions were supported by the results, but conduction other analysis as suggested above could change the discussion and conclusion. There is a paragraph in the discussion that is not relevant for this article (from line 383 to 410).

For example, there are few information available on other species than L. braziliensis and L. guyanensnsis. There is a possibility of LRV1-L.(Viannia) co-species evolution and maybe LRV from other species have a different role when interacting with the host. Furthermore, viral load was not well explored in clinical samples and this might have an important role.

**Editorial and Data Presentation Modifications?**

Reviewer #1: None, I would recomend publication in the way it is. English is very good and accurate. No further suggestions/corrections needed.

Reviewer #2: Results on Parasite load was not demonstrated and was presented very superficially. Raw data should be presented.

**Summary and General Comments**

Reviewer #1: This is a very interesting manuscript focused on the relationship of LRV and ML development. The paper is very well-written and all the procedures well performed according to their main hypothesis. I therefore recommend publication.

Reviewer #2: All comments were done above.

PLOS authors have the option to publish the peer review history of their article (what does this mean?). If published, this will include your full peer review and any attached files.

Reviewer #1: Yes: Rodrigo Pedro Soares

Reviewer #2: No

Figure Files:

Data Requirements:

Reproducibility:

References

---

## [Editor Report · Decision Letter 1]

11 Jan 2022

Dear Dr Valencia,

We are pleased to inform you that your manuscript 'Leishmania RNA virus-1 is similarly detected among metastatic and non-metastatic phenotypes in a prospective cohort of American Tegumentary Leishmaniasis' has been provisionally accepted for publication in PLOS Neglected Tropical Diseases.

Best regards,

Walderez O. Dutra, PhD.

Deputy Editor

Walderez Dutra

Deputy Editor

---

## [Editor Report · Acceptance letter]

24 Jan 2022

Dear Dr Valencia,

We are delighted to inform you that your manuscript, "Leishmania RNA virus-1 is similarly detected among metastatic and non-metastatic phenotypes in a prospective cohort of American Tegumentary Leishmaniasis," has been formally accepted for publication in PLOS Neglected Tropical Diseases.

Best regards,

Shaden Kamhawi

co-Editor-in-Chief

Paul Brindley

co-Editor-in-Chief
